



# Future changes in runoff over western and central Europe: disentangling the hydrological behavior of CMIP6 models

Juliette Deman[1], Julien Boé[1]

[1] CECI, Université de Toulouse, CERFACS/CNRS, Toulouse, France

*Correspondence to*: Juliette Deman (deman@cerfacs.fr)

**Abstract.** A large ensemble of climate projections from the Coupled Model Intercomparison Project Phase 6 is analyzed to characterize changes in runoff over western and central Europe in the late 21$^{st}$ century under a high-end emissions scenario. Our second objective is to gain a better understanding of the mechanisms responsible for the inter-model uncertainties. For this purpose, the models are grouped according to their hydrological response using a hierarchical classification algorithm.

Additional sensitivity experiments from two Model Intercomparison Projects are examined to better assess the role of the soil moisture-precipitation feedback and of the physiological impact of $CO_2$ in this context.

Half of the clusters show no significant change or a slight increase in annual runoff, while the others show a substantial decrease. Even when models agree on the annual changes in runoff, the changes in precipitation and evapotranspiration that drive them can be very different, even in terms of sign. Seasonal changes further differentiate the hydrological behavior of the

different clusters.

It is difficult to reject any cluster of models based on their accuracy in representing climatological averages and recent trends. The link between present-day averages or trends and future changes is generally weak and there are in general no major inconsistencies with reference datasets, partly because of large observational uncertainties.

Finally, we show that large-scale circulation and the representation of the physiological impact of $CO_2$ are important for the

extreme hydrological changes projected by some models. The soil-moisture precipitation feedback is important for the multi-model ensemble mean but not for the inter-model spread.



## 1 Introduction

The hydrological cycle is expected to change substantially with global warming and continued greenhouse gas emissions (Douville et al., 2021). Given the Clausius-Clapeyron relationship, the water-holding capacity of the atmosphere increases

with temperature. This, combined with the increasing amount of energy available at the Earth's surface, results in a greater evaporative demand (Scheff and Frierson, 2014). Constrained by the Earth's energy balance and partially offset by fast atmospheric adjustments, global precipitation is projected to increase at a rate of 2-3% per degree of global warming (Allan et al., 2020).

Different hydrological changes are expected in different regions, depending for example on current water balance, on the local

energy balance and on the influence of oceanic and/or atmospheric circulation on the regional moisture transport (Collins et al., 2013). In Europe, changes in atmospheric circulation are expected to have important impacts on precipitation changes, both in winter (e.g. Tuel and Eltahir, 2020), and in summer (e.g. Boé et al., 2008). More locally, land-atmosphere interactions are also expected to have important impacts on the hydrological cycle, especially in summer over a large part of Europe (Seneviratne et al., 2006, Boé and Terray, 2008). Because of the soil moisture-precipitation feedback (e.g. Koster et al., 2004;

Seneviratne et al., 2010) the depletion of soil moisture in winter and spring (Tuel and Eltahir, 2021) or in summer because of circulation-driven precipitation changes (Boé et al., 2008) may lead to an amplification of the summer precipitation decrease (Seneviratne et al., 2013). More recently, the physiological effect of $CO_2$ has emerged as a potentially important factor in hydrological changes (Lemordant et al., 2018), due its direct effect on evapotranspiration, and indirect effect on precipitation. Increasing atmospheric $CO_2$ concentrations can have a negative impact on plant transpiration through reduced stomatal

opening and/or reduced stomatal density, and a positive one through increased leaf area index, with the former impact being generally dominant (Lemordant et al., 2018).

These mechanisms lead to a projected increase in precipitation and evapotranspiration over Northern Europe (McKenna and Maycock, 2022), and a projected decrease over the Mediterranean area (Brogli et al., 2019). On an annual scale, runoff results from the balance between precipitation and evapotranspiration, and will therefore increase in regions where the increase in

precipitation is greater than the increase in evapotranspiration, but also in regions where the decrease in precipitation is smaller than the decrease in evapotranspiration. Runoff is projected to increase in northern Europe and to decrease in southern Europe (Zhao and Dai, 2022). In between these regions where the sign of runoff changes is robust, i.e. over western and central Europe (WCE), the sign of future runoff changes is particularly uncertain (Zhao and Dai, 2022).

These uncertainties in future runoff changes reflect the uncertainties associated with the mechanisms described above.

Projected changes in large-scale atmospheric circulation over the North Atlantic region are highly uncertain in climate models and are the cause of a large inter-model spread in CMIP5 climate projections (Shepherd, 2014). This uncertainty is related to the diversity of processes involved and to their representation in climate models (e.g. with respect to storm tracks, low frequency variability, blocking, Woollings, 2010). The physiological effect of $CO_2$ is known to cause a large inter-model spread in summer evapotranspiration changes over Europe (Boé, 2021), which is related to the complexity of its modeling and

important knowledge gaps (Vicente-Serrano et al., 2022). A large inter-model spread is also associated with the soil moisture-precipitation feedback (e.g. Seneviratne et al., 2013). In regions where evapotranspiration tends to be limited by soil moisture, such as WCE in summer in many models (Boé and Terray, 2008), drier soils lead to a decrease in evapotranspiration, which can lead to changes in precipitation, either directly through moisture recycling, or indirectly through changes in atmospheric stability. The magnitude of this feedback is not well known, and even its existence and / or sign is not clear across Europe

depending on the modeling framework (Hohenegger et al., 2009; Leutwyler et al., 2021, Lee and Hohenegger, 2024). In





addition to the uncertainties associated with land-atmosphere interactions, anthropogenic aerosols may also play a substantial role in the inter-model spread in summer evapotranspiration changes over Europe (Boé, 2016).

This study aims to better characterize hydrological changes over western and central Europe in the large ensemble of projections from the latest generation of global climate models from the Coupled Model Intercomparison Project (CMIP6, Eyring et al., 2016), and to assess their robustness. In this objective, particular attention is paid to evaluating how well the models represent the hydrological cycle in the current climate, in order to assess the credibility of their future projections. The inter-model differences in hydrological changes are investigated in details. Their causes are studied, building when possible on different sensitivity experiments realized within several Model Intercomparison Projects (MIP) from CMIP6.

The data, models and methods used in this study are described in Sect. 2. The projected changes in the hydrological cycle over WCE are then presented in Sect. 3. Model biases are evaluated in Sect. 4. In Sect. 5, the potential role of different mechanisms on the inter-model spread is assessed. Finally, the conclusions of this study are presented in Sect. 6.

## 2 Data and Methods

### 2.1 Data

An ensemble of 36 climate models participating in CMIP6 is analyzed, for the current climate over the period 1985-2014 and for future changes, between the periods 2081-2100 and 1995-2014 (Table 1). All available members differing by their initial conditions for both the historical simulations and the simulations forced by the Shared Socioeconomic Pathways SSP5-8.5 scenario (O'Neil et al., 2017, ssp585 simulations) are considered and referred to as the "ALL" experiments. The study of the SSP5-8.5 scenario at the end of the 21$^{st}$ century allows the hydrological responses to be maximized and facilitates the study of differences between models. Experiments from the Coupled Climate-Carbon Cycle Model Intercomparison Project (C4MIP, Jones et al., 2016), hist-bgc and ssp585-bgc, are also studied. These simulations are identical to historical and ssp585 simulations, except that the radiative effect of $CO_2$ is deactivated. These experiments are referred to as "BGC". Two simulations from the Land surface, Snow and Soil Moisture Model Intercomparison Project (LS3MIP, van den Hurk et al., 2016) are also analyzed. The amip-lfmip-rmLC and amip-lfmip-pdLC cover the 1980-2100 period and are forced by the SSP5-8.5 scenario after 2014 and the historical forcings before that. In both simulations, sea surface temperatures and sea ice concentrations are prescribed, from corresponding historical and ssp585 simulations. In the amip-lfmip-pdLC experiment, the land surface states are prescribed from the mean annual cycle over 1980-2014 of the corresponding historical Global Climate Model (GCM) simulations, while in the amip-lfmip-rmLC experiment, the land surface states are prescribed using a transient 30-year running mean from the historical and ssp585 simulations. Eight and seven models are available for the C4MIP and LS3MIP simulations analyzed in this work, respectively, with one member per model (Table 1). The variables considered in this study are precipitation (P), evapotranspiration (ETR), transpiration (Tran), potential evapotranspiration (PET), total runoff (R), surface soil moisture (SSM) and sea level pressure (SLP). Potential evapotranspiration is calculated for the 24 CMIP6 models with the necessary data (Table 1), for one member per model, according to the Penman-Monteith equation as defined by the United Nations Food and Agriculture Organization for deriving grass reference potential evapotranspiration (Pereira et al., 1999) with daily mean temperature, specific humidity, near-surface wind speed, incoming infrared and solar radiation at surface. Our analysis focuses on Europe, in particular on the western central Europe (WCE) region from the IPCC climate reference regions, as defined by Iturbide et al. (2020) and shown in Fig. 2.

Three observational datasets are used for the evaluation of historical simulations: ERA5-Land (Muñoz-Sabater et al., 2021), the Global Land Evaporation Amsterdam Model (GLEAM, Miralles et al. 2011; in prep) version 4.1a, and gridded monthly data over land from the Climatic Research Unit (CRU) time series (TS) version 4 (Harris et al., 2020). ERA5-Land consists



100  of the land component of the ERA5 reanalysis and is forced by ERA5 meteorological fields. Its enhanced resolution and increased complexity in land surface representation leads to an added value compared to ERA5 for the estimation of runoff and soil moisture, among other land surface variables (Muñoz-Sabater et al., 2021). GLEAM uses observations of surface net radiation, near-surface air temperature, wind speed, leaf area index and vapor pressure deficit to estimate potential evapotranspiration with the Penman's equation. The surface soil moisture from satellite observations is assimilated into the

105  soil profile. The potential evapotranspiration is then combined with an evaporative stress factor based on root-zone soil moisture. The evapotranspiration is finally computed as the sum of transpiration, interception, bare soil evaporation, evaporation for water bodies and evaporation for regions covered by ice and/or snow. Potential evapotranspiration and precipitation from CRU TS are also analyzed. CRU TS is based on the interpolation of weather station observations on a 0.5° resolution grid. The CRU TS potential evapotranspiration is computed with the Penman-Monteith formula from gridded mean

110  temperature, vapor pressure, cloud cover and climatological wind field values.

| Numbering | CMIP6 model | ECS | ALL (P, ETR, Tran, R, SLP) | ALL (SSM) | ALL (PET) | LS3MIP (P, ETR) | BGC (P, ETR, R) |
|---|---|---|---|---|---|---|---|
| 1 | ACCESS-CM2 | 4,72 | 10 | 5 | 1 | | |
| 2 | ACCESS-ESM1-5 | 3,87 | 40 | 40 | 1 | | 1 |
| 3 | BCC-CSM2-MR | 3,02 | 1 | 1 | 1 | | |
| 4 | CAMS-CSM1-0 | 2,29 | 1 | 1 | | | |
| 5 | CESM2 | 5,15 | 3 | 3 | 1 | 1 | |
| 6 | CESM2-WACCM | 4,68 | 1 | 3 | 1 | | |
| 7 | CanESM5-CanOE | | 3 | 3 | | | |
| 8 | CanESM5 | 5,64 | 25 | 25 | 1 | | 1 |
| 9 | CMCC-CM2-SR5 | 3,52 | 1 | 1 | 1 | | |
| 10 | CMCC-ESM2 | | 1 | 1 | 1 | 1 | |
| 11 | CNRM-CM6-1 | 4,90 | 6 | 6 | 1 | 1 | |
| 12 | CNRM-CM6-1-HR | 4,33 | 1 | 1 | | | |
| 13 | CNRM-ESM2-1 | 4,79 | 5 | 5 | | | 1 |
| 14 | E3SM-1-1-ECA | | 1 | 1 | | | 1 |
| 15 | E3SM-1-1 | | 1 | 1 | | | 1 |
| 16 | EC-Earth3 | 4,10 | 50 | | 1 | 1 | |
| 17 | EC-Earth3-Veg | 4,33 | 1 | 1 | | | |
| 18 | EC-Earth3-Veg-LR | | 1 | 1 | | | |
| | GFDL-CM4 | 3,89 | | | 1 | | |
| 19 | GFDL-ESM4 | | 1 | 1 | 1 | | |
| 20 | GISS-E2-1-G | 2,71 | 5 | 5 | | | 1 |
| 21 | GISS-E2-1-H | 3,12 | 5 | 5 | | | |
| 22 | HadGEM3-GC31-LL | 5,55 | 4 | | 1 | | |
| 23 | HadGEM3-GC31-MM | 5,42 | 4 | | 1 | | |
| 24 | INM-CM4-8 | 1,83 | 1 | | 1 | | |
| 25 | INM-CM5-0 | 1,92 | 1 | | 1 | | |
| 26 | IPSL-CM6A-LR | 4,56 | 7 | 7 | | 1 | |
| | KACE-1-0-G | 4,48 | | | 1 | | |
| 27 | MCM-UA-1-0 | 3,65 | 1 | | | | |



| | | | | | | |
|---|---|---|---|---|---|---|
| 28 | MIROC-ES2L | 2,66 | 10 | 10 | | |
| 29 | MIROC6 | 2,60 | 50 | 50 | 1 | 1 |
| 30 | MPI-ESM1-2-HR | 2,98 | 1 | 1 | 1 | |
| 31 | MPI-ESM1-2-LR | 3,00 | 50 | 30 | 1 | 1 |
| 32 | MRI-ESM2-0 | 3,13 | 5 | 5 | 1 | | 1 |
| 33 | NorESM2-LM | 2,56 | 1 | | 1 | | 1 |
| 34 | NorESM2-MM | 2,50 | 1 | | 1 | | 1 |
| 35 | TaiESM1 | 4,31 | 1 | 1 | | |
| 36 | UKESM1-0-LL | 5,36 | 5 | 5 | 1 | |
| | **Total** | | **305** | **219** | **24** | **7** | **8** |

**Table 1: CMIP6 models analyzed in this study. The left column shows the identification number assigned to each model in this work. The equilibrium climate sensitivity (ECS) of the models is given in the third column. The values are taken from Zelinka et al. (2020) and completed by Schlund et al. (2020) (in blue). The number of members used for each experiment is given in the next columns. For a given model, only the members differing by their initial conditions are considered. The total number of simulations per experiment is given in the last row.**

## 2.2 Classification of models

To facilitate the analysis of the CMIP6 multi-model ensemble, the hydrological responses of the 36 models studied in this work are classified using hierarchical clustering with Ward's linkage, similar to Monerie et al. (2016). The inter-cluster distances are calculated using Euclidean distance based on the seasonal relative changes in precipitation, evapotranspiration and runoff, averaged over WCE between the periods 2081-2100 and 1995-2014 for the SSP5-8.5 scenario. When multiple members are available, the multi-member ensemble mean is used. Relative changes for each season and variable are first standardized by subtracting the multi-model mean and dividing by the inter-model standard deviation. The classification algorithm starts with 36 clusters (one per model) and then, recursively, the two closest clusters are merged, until one cluster remains. To achieve the dual objective of effectively discriminating the hydrological behaviors and avoiding a large number of sparsely populated clusters, eight clusters are finally defined empirically (Fig. 1). The climate models that share several components (Table 2) are generally in the same cluster, except for NorESM2-LM/NorESM2-MM. This is somewhat surprising, as NorESM2-LM and NorESM2-MM differ only in resolution (and a very limited number of parameters in the atmosphere component, Seland et al. (2020)). Internal variability could explain why these two very similar models belong to different clusters, especially since only one member is available for these two models. The Earth System Models (ESMs) and Coupled Models (CM) developed by the same group (such as CNRM-CM6-1 and CNRM-ESM2-1, or CMCC-ESM2 and CMCC-CM2-SR5) are generally in the same cluster, suggesting a secondary role for the additional components included in these ESMs. Interestingly, the models from C6 all share the same land surface and/or atmosphere components (CLM and CAM, respectively), except for BCC-CSM2-MR. Two clusters, grouping only different versions of the same model, C7 (EC-Earth-Consortium models) and C8 (CCCma models), are quite far from the rest of the models, pointing to distinct hydrological behaviors.



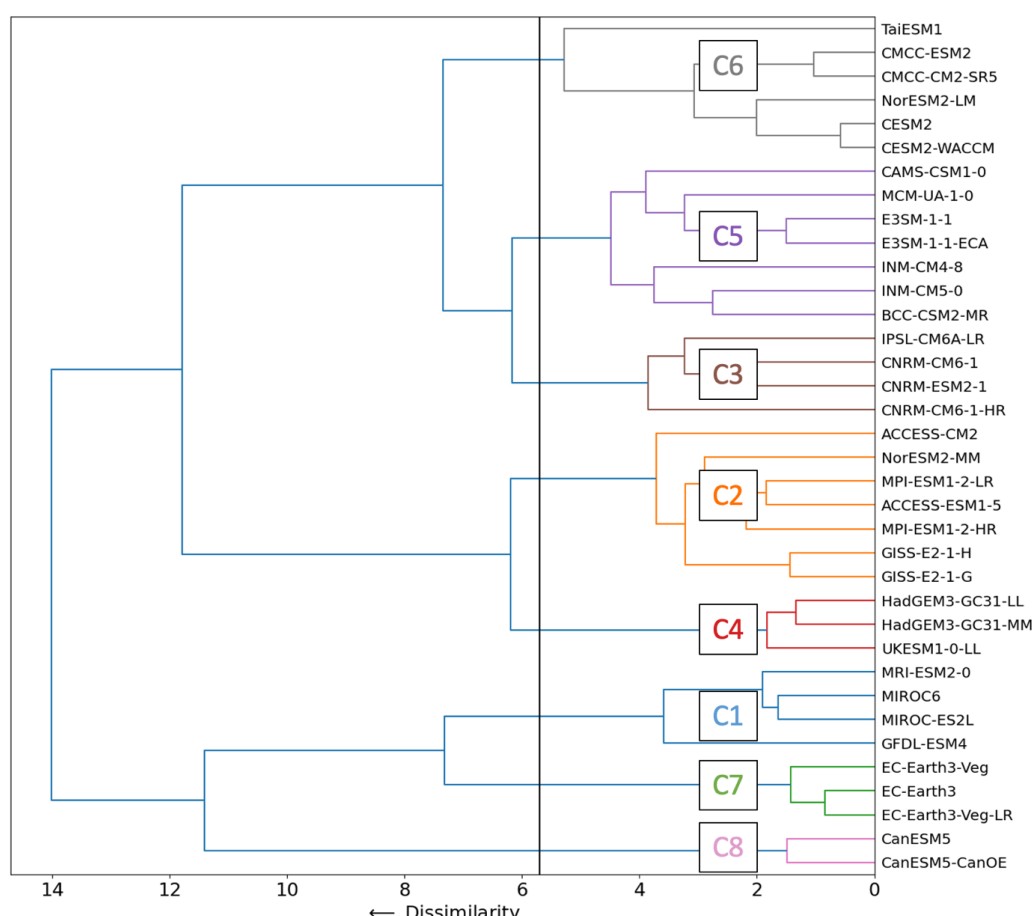

**Figure 1: Hierarchical clustering tree of the 36 CMIP6 models analyzed in this study based on their seasonal changes in precipitation, evapotranspiration and runoff, averaged over WCE between the periods 2081-2100 and 1995-2014. The clusters are labeled from C1 to C8. Models within the same cluster share the same color.**


| Cluster | CMIP6 model | Land Surface Component | Atmospheric component | Oceanic component |
|---------|-------------|------------------------|-----------------------|-------------------|
| C1 | MRI-ESM2-0 | HAL 1.0 | MRI-AGCM3.5 | MRI.COM4.4 |
| | MIROC6 | MATSIRO6.0 | CCSR AGCM | COCO4.9 |
| | MIROC-ES2L | MATSIRO6.0 + VISIT-e ver1.0 | CCSR AGCM | COCO4.9 |
| | GFDL-ESM4 | GFDL-LM4 | GFDL-AM4.1 | GFDL-OM4p5 (MOM6) |
| C7 | EC-Earth3-Veg | LPJ-GUESS | IFS 36r4 | NEMO3.6 |
| | EC-Earth3 | HT-ESSEL | IFS 36r4 | NEMO3.6 |
| | EC-Earth3-Veg-LR | LPJ-GUESS | IFS 36r4 | NEMO3.6 |
| C2 | ACCESS-CM2 | CABLE2.5 | MetUM-HadGEM3-GA7.1 | ACCESS-OM2 (GFDL-MOM5) |
| | NorESM2-MM | CLM | CAM-OSLO | MICOM |
| | MPI-ESM1-2-LR | JSBACH3.20 | ECHAM6.3 | MPIOM1.63 |
| | ACCESS-ESM1-5 | CABLE2.4 | HadGAM2 | ACCESS-OM2 (GFDL-MOM5) |
| | MPI-ESM1-2-HR | JSBACH3.20 | ECHAM6.3 | MPIOM1.63 |
| | GISS-E2-1-H | GISS LSM | GISS-E2.1 | GISS HYCOM |



| | GISS-E2-1-G | GISS LSM | GISS-E2.1 | GISS OCEAN |
|---|---|---|---|---|
| C4 | HadGEM3-GC31-LL | JULES-HadGEM3-LG7.1 | MetUM-HadGEM3-GA7.1 | NEMO-HadGEM3-GO6.0 |
| | HadGEM3-GC31-MM | JULES-HadGEM3-LG7.1 | MetUM-HadGEM3-GA7.1 | NEMO-HadGEM3-GO6.0 |
| | UKESM1-0-LL | JULES-ES-1.0 | MetUM-HadGEM3-GA7.1 | NEMO-HadGEM3-GO6.0 |
| C3 | IPSL-CM6A-LR | ORCHIDEE | LMDZ | NEMO-OPA |
| | CNRM-CM6-1 | Surfex 8.0c | Arpege6.3 | NEMO3.6 |
| | CNRM-ESM2-1 | Surfex 8.0c | Arpege6.3 | NEMO3.6 |
| | CNRM-CM6-1-HR | Surfex 8.0c | Arpege6.3 | NEMO3.6 |
| C6 | TaiESM1 | CLM4.0 | TaiAM1 | POP2 |
| | CMCC-ESM2 | CLM4.5 | CAM5.3 | NEMO3.6 |
| | CMCC-CM2-SR5 | CLM4.5 | CAM5.3 | NEMO3.6 |
| | NorESM2-LM | CLM | CAM-OSLO | MICOM |
| | CESM2 | CLM5 | CAM6 | POP2 |
| | CESM2-WACCM | CLM5 | WACCM6 | POP2 |
| C5 | CAMS-CSM1-0 | CoLM 1.0 | ECHAM5_CAM5 | MOM4 |
| | MCM-UA-1-0 | Standard Manabe Bucket hydrology scheme (1969) | R30L14 | MOM1.0 |
| | E3SM-1-1 | ELM (based on CLM4.5) | EAM (based on CAM5) | MPAS-Ocean |
| | E3SM-1-1-ECA | ELM (based on CLM4.5) | EAM (based on CAM5) | MPAS-Ocean |
| | INM-CM5-0 | INM-LND1 | INM-AM5-0 | INM-OM5 |
| | INM-CM4-8 | INM-LND1 | INM-AM4.8 | INM-OM5 |
| | BCC-CSM2-MR | BCC-AVIM2 | BCC_AGCM3_MR | MOM4 |
| C8 | CanESM5 | CLASS3.6 and CTEM1.2 | CanAM5 | Nemo3.4.1 |
| | CanESM5-CanOE | CLASS3.6 and CTEM1.2 | CanAM5 | Nemo3.4.1 |

**Table 2: Models classified in this study, sorted by cluster and name of their main components for the land surface, the atmosphere, and the ocean (from Notz and Kern, 2024). The colors of the first column correspond to the clusters, and are the same as those used in Fig. 1 and throughout the paper when presenting the results of the different clusters.**

## 3 Future Changes

Because of the large uncertainties involved, this study focuses mainly on changes in the hydrological cycle over the WCE region (Fig. 2). Consistent with the results of Zhao and Dai (2022), changes in precipitation, evapotranspiration and runoff are indeed generally robust over Northern Europe and the Mediterranean (Fig. 2). The three variables increase in the former region and decrease in the latter. The changes in surface soil moisture are more spatially homogeneous over Europe, with a general decrease. Changes in the hydrological cycle are more uncertain over WCE. Multi-model mean changes in precipitation are

weak and generally do not emerge from internal variability. Multi-model mean changes in evapotranspiration are also generally weak except over the Alps and the north of WCE. Changes in evapotranspiration emerge from internal variability in more than two third of models, but models do not agree on their sign, pointing to fundamental modelling uncertainties. The runoff generally decreases over WCE, but the changes generally do not emerge from internal variability.





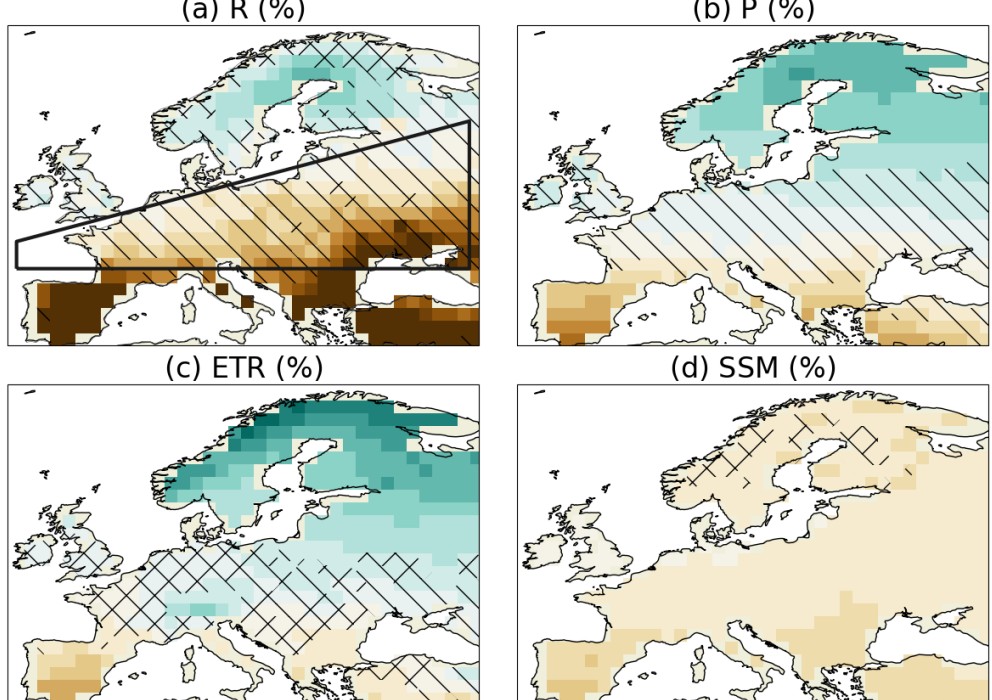

**Figure 2: Multi-model ensemble mean of annual relative changes (%) between 1995-2014 and 2081-2100 under the SSP5-8.5 scenario**
**in (a) runoff, (b) precipitation, (c) evapotranspiration and (d) surface soil moisture. 36 CMIP6 models are used (see Table 1). On**
**these maps, one member per model is used. The hatched area shows where less than 66% of the models project changes greater than**
**the variability threshold. The crossed area shows where more than 66% of the models project changes greater than the variability**
**threshold and less than 80% of the models agree on the sign of the changes. The variability threshold is defined as $\gamma = \sqrt{2/20} *$**
**$1.645 * \delta_{1yr}$ where $\delta_{1yr}$ is the interannual standard deviation calculated in a linearly detrended modern period (Gutiérrez et al.,**
**2021; Zappa et al., 2021). The enclosed area in (a) is the western and central Europe (WCE) region, as defined in the Sixth Assessment**
**Report (Iturbide et al., 2020).**

In order to better assess the inter-model differences, the seasonal and annual changes in runoff in WCE for individual CMIP6
models and all available members are shown in Fig. 3. Strong negative changes in annual runoff are projected by 15 models
(Fig. 3a), with the amplitude of the absolute changes exceeding the variability threshold (with the same definition as in Fig. 2)
for most members. Five models project a significant increase in runoff. Summer runoff decreases in 31 out of 36 models, and
this decrease is greater than the variability threshold in 24 models. Only three models show a significant increase in summer
runoff. Most models also agree on a decrease in runoff in fall (Fig. 3b). A significant decrease in runoff is also the dominant
response in spring, although it occurs in less than 50% of models (Fig. 3d). Only in winter is a significant increase in runoff
the dominant response (Fig. 3c). The impact of internal variability on future changes is large, as shown by the models with a
substantial number of members. The spread due to internal variability can be as large as 0.35 mm day[-1] for some models in
winter and spring (Fig. 3c, d).



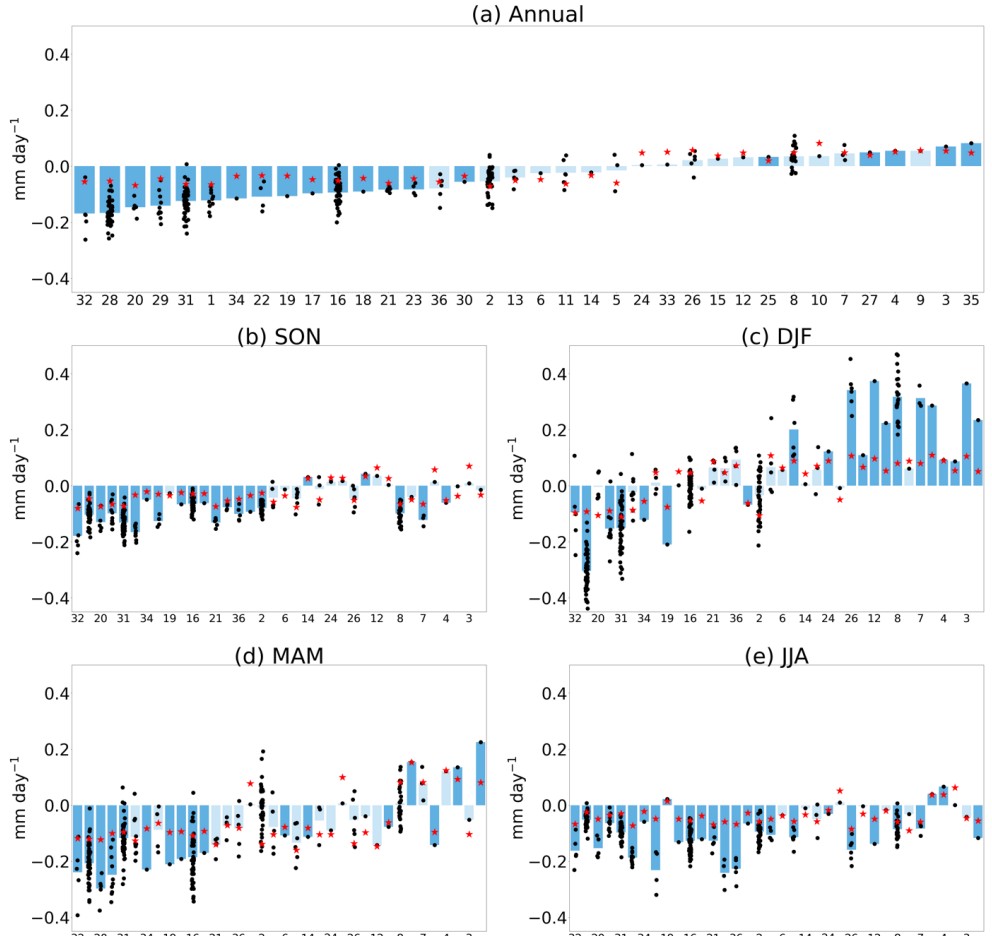

**Figure 3: Changes (mm day$^{-1}$) in (a) annual and seasonal ((b) autumn, (c) winter, (d) spring and (e) summer) runoff over WCE,**
**between 1995-2014 and 2081-2100 under the SSP5-8.5 scenario. Each bar corresponds to the multi-member mean of the model whose**
**identification number (see Table 1) is given on the x-axis. The models are ordered by ascending value of annual runoff changes. The**
**dots correspond to the changes of each member. The red stars correspond to the variability threshold, defined as $\gamma = \sqrt{2/20} *$**
**$1.645 * \delta_{1yr}$ where $\delta_{1yr}$ is the interannual standard deviation calculated in a linearly detrended modern period (Gutiérrez et al.,**
**2021; Zappa et al., 2021). The variability threshold is multiplied by -1 when the multi-member mean is negative. Dark blue indicates**
**that the changes in more than 66% of members exceed the multi-member mean of the variability threshold.**

We now explore the role of changes in precipitation and evapotranspiration, the main drivers of runoff, on these differences
in runoff changes, based on the model classification introduced in Section 2b. The eight clusters mostly show two different
responses in annual runoff changes (Fig. 4a). Four clusters (C1, C7, C2, C4) show a decrease in runoff ranging from -10% to
-25% and four clusters (C3, C6, C5, C8) show no change or a slight increase in runoff, ranging from -5% to +10%.
Interestingly, similar changes in runoff can be caused by very different changes in precipitation and evapotranspiration (Fig.
4b, c), and the sign of runoff changes is not always determined by the sign of precipitation changes. Changes in runoff are
indeed negative in C1 and C7, even if precipitation increases, because evapotranspiration also increases. Conversely, in C2
and C4, the decrease in runoff is associated with a decrease in precipitation, as changes in evapotranspiration are small and
even negative in C4. C8 (CCCma models) shows a small increase in runoff despite a very strong increase in precipitation





because evapotranspiration also strongly increases. C5, C3 and C6 show no changes or small increases in precipitation and evapotranspiration, leading to small positive changes in runoff or no changes.

The cluster analysis thus shows that different mechanisms can lead to a similar response in annual runoff, and that the multi-model mean is uninformative, and even misleading, about annual changes in runoff. The multi-model mean for changes in annual runoff indeed falls between the two groups of clusters, and is therefore not representative of either (gold line in Fig.

4a).

Some clusters also show very specific behaviors for seasonal changes. The large annual increases in precipitation and evapotranspiration of C8 are explained by strong positive changes in winter and spring (Fig. 4h, k) not compensated by decreases in summer (Fig. 4, n). The EC-Earth-Consortium models (C7) also project a large increase in precipitation and evapotranspiration in winter and spring (Fig. 4h, i, k, l). At the other end of the spectrum, C4 (MOHC models) shows strong

decreases in summer precipitation and evapotranspiration compared to the other clusters.

The eight clusters obtained by hierarchical clustering and analyzed in this section represent very different possible future evolutions of the hydrological cycle over WCE. In the next section, we assess whether they also behave differently in the present climate, in terms of climatological means and trends, and whether some clusters could be disqualified due to inconsistencies with observational estimates.





**Figure 4:** Relative changes (%) between 1995-2014 and 2081-2100 under the SSP5-8.5 scenario, in (a, d, g, j, m) runoff, (b, e, h, k, n) precipitation, and (c, f ,l, l, o) evapotranspiration for the eight clusters identified through hierarchical clustering (see Section 2), for annual (a, b, c), autumn (d, e, f), winter (g, h, i), spring (j, k, l) and summer (m, n, o) changes. The clusters are arranged in ascending order based on their mean annual changes in runoff. The gold dashed line corresponds to the multi-model mean. The 5-95% confidence interval is shown with error bars for each cluster, and the results of individual models are shown with dots whose color corresponds to the cluster.



## 4 Present-day evaluation

The climatological means and recent trends of various hydrological variables for the eight clusters are now calculated and compared with reanalyzes and observational estimates (see Section 2).

The realism of the model for mean precipitation over 1985-2014 is difficult to assess due to the large differences between the two reference datasets (Fig. 5a). Most models are within or close to the range of the two reference datasets. There is also a large observational uncertainty in climatological potential evapotranspiration (Fig. 5c). It is overestimated in many CMIP6 models compared to CRU TS and underestimated compared to GLEAM (Fig. 5c). Models in C6 and C5 show large values of climatological potential evapotranspiration. This doesn't lead to large climatological values of evapotranspiration (Fig. 5e),

probably because C6 and C5 models are also characterized by relatively low climatological precipitation and therefore less water at surface for evapotranspiration. On the contrary, C1 models are generally characterized by high evapotranspiration despite relatively low potential evapotranspiration, probably due to high climatological precipitation. Transpiration is severely underestimated by all CMIP6 models compared to GLEAM, especially the CCCma models (C8), with the exception of two models in C5 (E3SM models, Fig. 5g). The CMIP6 models are generally close to the climatological runoff from ERA5 land,

except for one outlier model in C3. (Fig. 5i).

Except for one model from C7, none of the CMIP6 models show a significant trend in precipitation, which is consistent with the two observational estimates. The trends in potential evapotranspiration are positive and significant in both reference datasets. This is expected in the context of climate change, with an increase in the energy available at the surface associated with greenhouse gases, and also probably during this period, with global brightening (Wild, 2009).

The trends in evapotranspiration in both reference datasets are significantly positive, probably driven by the trends in potential evapotranspiration. Most CMIP6 models also simulate positive trends, but they are not always significant. Models from C7 (EC-Earth-Consortium models), which project strong future increases in evapotranspiration (Fig. 4), show the strongest present-day trends in evapotranspiration.

The simulated trends in transpiration are significantly positive in models from C1, C7, C5 and C8, in line with the significant

positive trend in GLEAM. The other clusters show smaller or even negative non-significant trends in transpiration. As with total evapotranspiration, the EC-Earth-Consortium models (C7) show particularly strong positive trends in transpiration. The simulated trends in runoff are negative and not significant in most models as in ERA5-land.

Based on the evaluation discussed in this section, it is difficult to reject the future projections of specific clusters on the basis on of how well they compare with observations in the present climate. Indeed, their performance varies depending the variable

and evaluation metric. In addition, limitations and uncertainties in reference datasets are important. They significantly hinder the evaluation of the continental hydrological cycle in climate models. Many hydrological variables are poorly observed, making it necessary to use models, constrained more or less strongly by observations (such as GLEAM and ERA5-land). The results of the evaluation described in this section must therefore be treated with caution.

A large and quasi-generalized underestimation of climatological transpiration in climate models compared to GLEAM is

found. The realism of GLEAM transpiration is difficult to assess due to the lack of direct observations with a good spatial coverage. However, the underestimation of transpiration noted here is consistent with the results of Lian et al. (2018), who used sparse isotopic and non-isotopic measurements to constrain the transpiration simulated by CMIP5 models. This underestimation could have implications for the future response of the models (Berg and Sheffield, 2019). Indeed, transpiration can use water from deeper reservoirs in the soil compared to bare soil evaporation. Moreover, transpiration depends on the

stomatal conductance of the plant, which could be affected by $CO_2$ through its physiological effect.





The analyses discussed in this section reveal interesting behaviors in some of the CMIP6 models that are useful for interpreting their future responses. The models with future decreases in annual runoff (C1, C7, C2 and C4, see Fig. 4a) tend to have higher present-day climatological values of evapotranspiration. The C7 models (EC-Earth models) have stronger trends in precipitation, evapotranspiration and transpiration compared to the other models (and to the observational estimates), which is consistent, for precipitation and evapotranspiration, with the stronger changes projected between 2081-2100 and 1995-2014 as noted in Fig. 4a, b.

In the next section, we go deeper in the understanding of the differences between models, focusing on some mechanisms that are known to be important for hydrological changes over Europe.





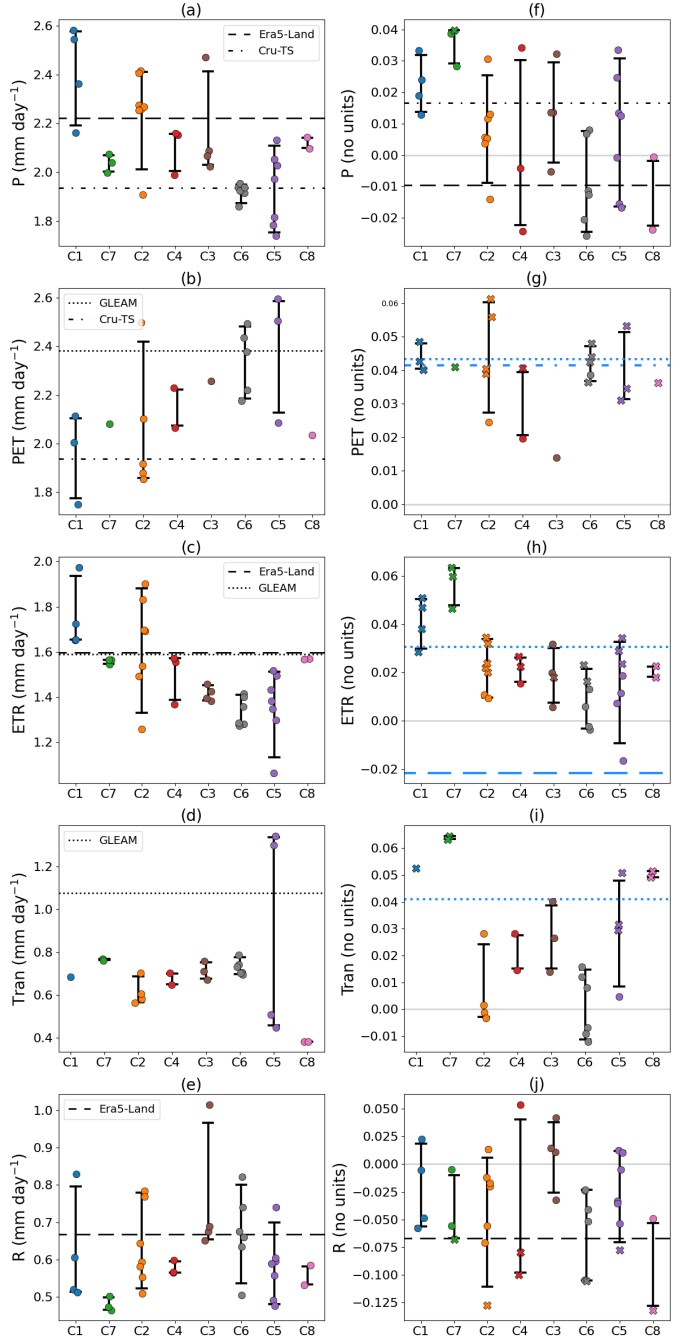

**Figure 5: Annual climatological mean (mm day⁻¹) over 1985-2014 of (a) precipitation, (c) potential evapotranspiration, (e) evapotranspiration, (g) transpiration and (i) runoff. Annual trends over 1985-2014 (relative to 1995-2014, no units) of (b) precipitation, (d) potential evapotranspiration, (f) evapotranspiration, (h) transpiration and (j) runoff. The horizontal lines correspond to reference datasets. Blue lines indicate that the trend is significant (p-value<0.01). The models are grouped based on the hierarchical clustering. The clusters are defined in Fig. 1. The distribution is displayed with error bars indicating the 5-95% confidence interval. The colored dots and cross correspond to individual models with a single member. The crosses indicate that the trend of the model is significant (p-value<0.01).**



## 5 Mechanisms

A salient feature of the cluster's hydrological response to climate change is the very strong increase in annual and winter precipitation in C8, and in one model of the C6 cluster, TaiESM1 (Fig. 4h). Given the potentially important role of large-scale
circulation in precipitation changes in winter (Shepherd, 2014), we assess the role of large-scale circulation in this context. The models generally project a strong positive sea level pressure anomaly over the Mediterranean in winter (Fig. 6a). TaiESM1 and the C8 models project opposite changes, with a strong negative anomaly over most of central Europe and northern Europe (Fig. 6b). This is important because there is a relationship between changes in sea level pressure and changes in precipitation over WCE in the CMIP6 models (Fig. 6c). Lower sea level pressures there lead to stronger westerlies, which in turn lead to
more precipitation over Europe in winter. The unusual sea level pressure changes of the C8 models and TaiESM1 probably explain to a substantial extent why these models project a very strong increase in winter precipitation over WCE compared to the rest of the models. Note however that TaiESM1 and the C8 models are even further away from the other models in terms of precipitation changes than in terms of sea level pressure changes (Fig. 6c), suggesting that other processes are also important.

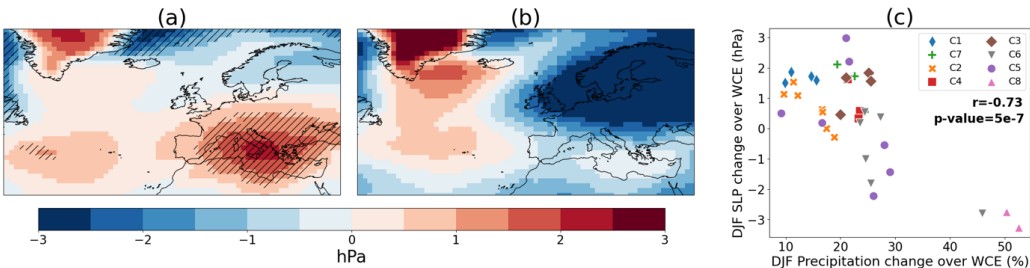

**Figure 6: Sea level pressure changes in winter (DJF), between 1995-2014 and 2081-2100 under the SSP5-8.5 scenario, for (a) the multi-model ensemble mean of all models except models from C8 and TaiESM1 and (b) multi-model ensemble mean of C8 models and TaiESM1. The hatched area in (a) indicates where at least 80% of models agree on the sign of the change. (c) Scatterplot between winter changes in precipitation (%) averaged over WCE (x-axis) and winter changes in sea level pressure (hPa) averaged over WCE (y-axis) of all CMIP6 models. The Pearson correlation coefficient ("r") and the corresponding p-value is given in the figure.**

Land atmosphere interactions are known to be important for changes in the European climate during summer, but their role on changes in the hydrological cycle and especially in the inter-model spread remains unclear. Simulations done within the LS3MIP project (van den Hurk et al., 2016; see Section 2) are useful in this context.

In the simulations with imposed present-day soil moisture (blue line), summer evapotranspiration increases throughout the 21st century (Fig. 7a), with a small increase in precipitation peaking in 2030, followed by a small negative trend until the end of
the 21st century (Fig. 7b). In the simulations with time-evolving soil moisture imposed from the historical and SSP5-8.5 simulations, summer evapotranspiration peaks in 2020 and stays roughly constant during the rest of the 21st century (Fig. 7a). This is consistent with the decrease in summer soil moisture in these simulations (not shown), more generally seen in the full ensemble of CMIP6 models (e.g. Fig. 2). Precipitation decreases much more in the simulations with imposed time-evolving soil moisture than in the simulations with imposed present-day soil moisture, consistent with the differences in
evapotranspiration changes. This analysis therefore shows how the decrease in soil moisture can lead to a decrease in precipitation through its impact on evapotranspiration. What we are more interested in is the role of the soil moisture feedback in the inter-model spread in future changes in evapotranspiration and precipitation. The strength of this feedback is characterized here as the difference of evapotranspiration or precipitation changes between the LS3MIP simulations forced by time-varying and constant present-day soil moisture, normalized by the corresponding difference in soil moisture changes. No
strong inter-model relationship is seen between future changes in evapotranspiration and precipitation in standard CMIP6 projections and the strength of the feedback (Fig. 8). The soil moisture-precipitation feedback thus does not seem to play an





important role in the inter-model spread in hydrological changes, and cannot explain the different behaviors of the clusters seen in Fig. 4n, o, at least based on the results of the models participating to LS3MIP.

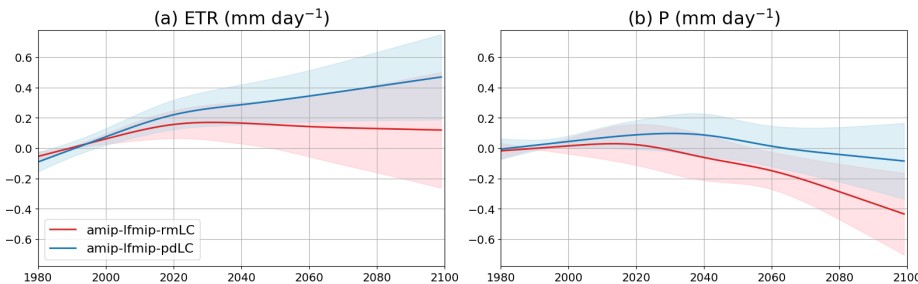

**Figure 7: Anomalies of summer (a) evapotranspiration (mm day⁻¹) and (b) precipitation (mm day⁻¹), averaged over WCE for the period 1980-2100 using the 1980-2000 period as reference, for two experiments from LS3MIP where soil moisture is prescribed. The red line shows the multi-model mean of the simulations where the running mean of soil moisture from the historical + SSP5-8.5 scenario is prescribed (amip-lfmip-rmLC). The blue line shows the multi-model mean of the simulations where the present-day climatological soil moisture is prescribed through the simulation (amip-lfmip-pdLC, see Section 2). The shaded area represents the**
**mean +/- one inter-model standard deviation. Seven models are used for this analysis (see Table 1).**

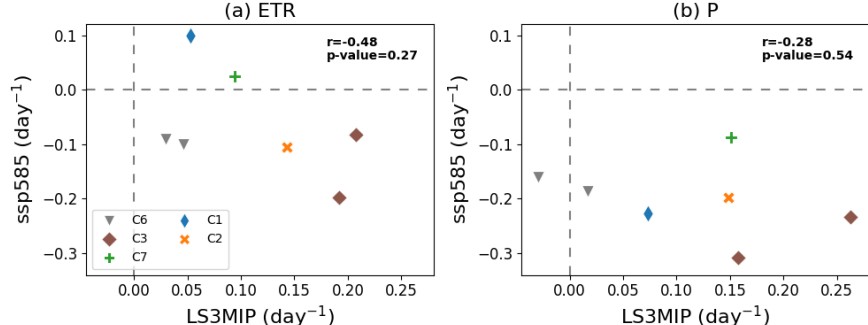

**Figure 8: (y-axis) Absolute changes in summer (a) evapotranspiration and (b) precipitation over WCE between 1995-2014 and 2081-2100 for the SSP5-8.5 scenario and historical simulations standardized by the changes in soil moisture in each model (day⁻¹) versus (x-axis) difference between changes in amip-lfmip-rmLC (evolving soil moisture) and amip-lfmip-pdLC (constant soil moisture),**
**standardized by the changes in soil moisture in the amip-lfmip-rmLC simulation in each model (day⁻¹). The Pearson correlation coefficients ("r") and corresponding p-values are shown in the figure. The seven models for which LS3MIP simulations are available are shown (Table 1).**

Finally, we assess whether the physiological effect of $CO_2$ can explain some of the inter-model spread in the future changes in the hydrological cycle over WCE in summer, based on simulations from C4MIP (see Section
2). Fig. 9 compares the future changes in evapotranspiration, precipitation and runoff from standard projections with the SSP5-8.5 scenario (with respect to historical simulations, "ALL") and the changes projected by identical simulations, except that the radiative effect of $CO_2$ is deactivated (hist-bgc and ssp585-bgc from C4MIP, "BGC"). Note that anthropogenic aerosols, among other forcings, still evolve in BGC. Therefore, the future changes projected in BGC should not be interpreted as resulting solely from the biogeochemical effect of $CO_2$. Significant inter-model correlations exist between the future changes
projected in ALL and BGC, for evapotranspiration, precipitation and runoff in summer. The largest correlation is obtained for evapotranspiration and the smallest one is obtained for runoff. UKESM1-0-LL (in C4) and CanESM5 (in C8) project extreme and opposite annual and summer changes in precipitation and evapotranspiration among the CMIP6 models. The CCCma models (C8) project no changes in precipitation and an increase in evapotranspiration during summer, while the MOHC models



(C4) project a very large decrease in precipitation and evapotranspiration (Fig. 4n, o). This is also true for BGC simulations,
which do not include the radiative effect of $CO_2$ (Fig. 9). This suggests that the radiative effect of $CO_2$ is not primarily
responsible for the extreme responses of these models in summer precipitation and evapotranspiration. The biogeochemical
effect of $CO_2$, and possibly other forcings such as aerosols also included in BGC simulations, could therefore account for a
substantial part of the inter-model spread in the changes of the hydrological cycle projected by the CMIP6 models in summer.

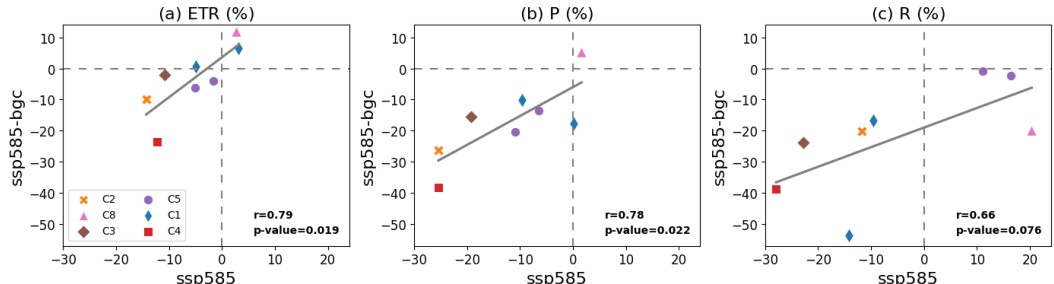

**Figure 9: Relative changes (%) between 1995-2014 and 2081-2100 in summer (a) precipitation, (b) evapotranspiration and (c) runoff,
averaged over WCE in the (y-axis) standard historical + ssp585 simulations compared to the (x-axis) BGC simulations for the eight
models with BGC simulations (Table 1). The linear regression line is shown in gray. The respective Pearson correlation coefficients
and corresponding p-values are given in the sub-figures.**

**6 Discussion and Conclusion**

This study has examined the projected changes in runoff over Europe at the end of the 21st century under a high emissions
scenario (SSP5-8.5) in a large ensemble of global climate models, as well as the changes in precipitation and evapotranspiration
that drive them.

Changes in runoff are particularly uncertain over western and central Europe, due to large uncertainties in changes in
precipitation and evapotranspiration. Depending on the model, significant increases, decreases or no significant changes in
annual runoff are possible. The most robust signal is seen in summer, with a significant decrease in runoff projected by the
vast majority of models.

To investigate the reasons for this large inter-model spread in future hydrological changes, the models have been first grouped
into clusters according to their seasonal changes in runoff, precipitation and evapotranspiration using hierarchical
classification. Climate models from the same modeling group are almost always clustered together, even when the models
differ in their modeling of the land surface. Additionally, a cluster regroups mainly models from different modeling groups
that share the same atmosphere and/or land surface components (CLM and CAM). This indicates, not surprisingly, that the
effective number of independent models regarding hydrological changes over WCE is much smaller than the total number of
CMIP6 models. Ensemble metrics such as the multi-model mean and inter-model standard deviation based on model
democracy could therefore be biased. This result also highlights the interest of thinking in terms of a few groups of models
with similar responses, depicting different possible storylines of future hydrological changes, as in this study.

Half of the clusters show no significant evolution or a slight increase in annual runoff, while the others show a substantial
decrease. Even in some clusters agree on the annual changes in runoff, the changes in precipitation and evapotranspiration that
drive them can strongly differ, even in terms of sign. The models that project a decrease in annual precipitation almost always
project a decrease in annual runoff, but several models show a decrease in runoff despite an increase in precipitation. These
models are characterized by a comparatively large increase in evapotranspiration. Seasonal changes further differentiate the
hydrological behaviors of the different clusters.





The present-day evaluation of the models does not allow the future projections of some clusters to be ruled out. In general, there is little relationship between the behavior of clusters with respect to future changes and their behavior with respect to current climatological averages or trends. A possible exception is that the models that project a decrease in runoff are also
generally characterized by higher climatological evapotranspiration. Among these models, the EC-Earth-Consortium models also show strong positive trends in evapotranspiration and precipitation, which are consistent with the strong positive changes that they project for the late 21$^{st}$ century. In addition, it is often difficult to assess the realism of the different clusters due to some large differences between the reference datasets used for the evaluation and/or intrinsic limitations. With this limitation in mind, there are no general and major inconsistencies between the CMIP6 multi-model ensemble and the observational
datasets, except that almost all models could strongly underestimate transpiration. However, the reference estimate must be treated with caution as it is not based on direct observations.

In order to better understand the large inter-model spread in hydrological changes over WCE, we have investigated several potentially important mechanisms in this context, drawing where possible on sensitivity experiments conducted in specific CMIP6 MIPs.
The large inter-model spread in the amplitude of winter precipitation changes over WCE is partly explained by large-scale circulation, and in particular sea level pressure changes over central and northern Europe. TaiESM1 and the CCCma models, which show a very large increase in winter precipitation over WCE, show unusual circulation changes.
We have shown that the soil moisture-precipitation feedback, estimated thanks to LS3MIP experiments, is an important process in shaping the ensemble mean changes in precipitation and evapotranspiration over WCE in summer, leading to substantial
drying. However, it has little influence on the inter-model spread in hydrological changes, at least for the small sample of models considered. It is important to note that the positive soil moisture-precipitation feedback suggested by this analysis might not be realistic given new results based on a short storm-resolving simulation (Lee and Hohenegger, 2024).
The analysis of experiments from C4MIP suggests that a substantial part of the large inter-model spread in summer hydrological changes over WCE could be attributed to the physiological effect of $CO_2$, possibly combined with the effect of
anthropogenic aerosols.

Overall, three groups of models (from the EC-Earth-Consortium, CCCma and MOHC modelling group) often show atypical hydrological behaviors, although their annual changes in runoff are not unusual. MOHC models are characterized by a strong physiological effect of $CO_2$ in summer, consistent with the very large decrease in both precipitation and evapotranspiration that they project during this season. The CCCma models are characterized by a very strong increase in precipitation, especially
in winter, which is partly related to unusual changes in large-scale circulation. These models also project large increases in evapotranspiration, consistent with a small physiological effect of $CO_2$. Interestingly, their climatological transpiration is small, which could limit the potential impact of the physiological effect of $CO_2$. The increase in evapotranspiration in CCCma models exceeds the increase in precipitation in summer and autumn, and the opposite is true in winter. In the end, these models project both a very large increase in runoff in winter, and a very large decrease in summer despite virtually no change in
precipitation in summer. The EC-Earth-Consortium models are also characterized by large increases in precipitation and even larger increases in evapotranspiration. In these models, the changes in evapotranspiration dominate, leading to decreases in runoff in all seasons except winter, where little change is projected. These models also simulate the largest present-day trends in annual evapotranspiration (despite average trends in potential evapotranspiration).

Note that the CCCma, MOHC and EC-Earth-Consortium models (as other models) are outside the likely range of equilibrium
climate sensitivity (ECS, Table 1) of 2.5-4K from the latest IPCC report (IPCC, 2021), and even outside the very likely range of 2-5K for the CCCma and MOHC models. Selecting only models within the likely range of ECS would therefore lead to the





exclusion of models with atypical hydrological behaviors over WCE. The pros and cons would have to be seriously weighed before making such a decision.

Uncertainties in the changes in the hydrological cycle over WCE are very large. Reducing the uncertainties is therefore critical
from an adaptation perspective. The study highlights the diverse mechanisms involved, making it challenging to identify a single appropriate metric for process-based observational constraint (Hegerl et al., 2021). The lack of direct observations with correct spatio-temporal coverage for many hydrological variables, and the large uncertainties in available hybrid model / observations estimates, also complicate the use of observational constraints based on past changes (e.g. Ribes et al., 2022). The strong model uncertainties in the changes arise in large part from the choice of the high-end scenario, which facilitates the
characterization of the differences in projected hydrological changes between models and the associated mechanisms. From a strict impact perspective, it would probably be more useful to use a more realistic scenario (Hausfather and Peters, 2020).

This study also shows the importance of the two-way coupling between changes in precipitation and evapotranspiration. Apart from the obvious influence of precipitation on evapotranspiration by modulating surface water availability, evapotranspiration can influence precipitation because of the soil moisture-precipitation feedback, acting independently or in conjunction with
the physiological effect of $CO_2$. This could be important, because impact studies are typically based on offline hydrological simulations decouple the changes in precipitation (coming from climate models) from the changes in evapotranspiration (calculated by the hydrological model). For example, forcing a hydrological model with a weak response in evapotranspiration, with data from a climate model with a large decrease in precipitation partly driven by a large decrease in evapotranspiration could lead to a strong and unrealistic decrease in runoff. The hydrological models, which generally require downscaled and
bias-corrected data, provide a finer representation of the hydrological cycle at the catchment-scale. The two approaches are therefore complementary.

This work is mainly focused on the inter-model spread in hydrological changes. Even on the late period and with the extreme scenario studied in this work to better isolate anthropogenic signals and associated mechanisms, a substantial impact of internal variability is noted. For shorter lead times, often more relevant from an adaptation and policy-making perspective, uncertainties
related to internal variability would be even more important and critical to study (e.g. Mankin et al., 2020).

**Data Availability**. The CMIP6 global climate projections are publicly available through the Earth System Grid Federation (https://esgf-node.ipsl.upmc.fr/search/cmip6-ipsl/)
The ERA5-Land data are available at https://cds.climate.copernicus.eu/cdsapp#!/dataset/reanalysis-era5-land
The Gleam data (version 4.1a) at https://www.gleam.eu/
And the CRU-TS data at https://crudata.uea.ac.uk/cru/data/hrg/cru_ts_4.07/

**Author Contribution.** J. D and J. B designed the study. Analyses were performed by J. D. The first draft of the manuscript was written by J. D, J. B and J. D commented on previous versions of the manuscript. All authors read and approved the final manuscript.

**Competing Interests.** The authors declare that they have no conflict of interest.

**Acknowledgments.** We acknowledge the World Climate Research Programme's Working Group on Coupled Modelling responsible panel for CMIP6. We also thank the climate modelling groups for producing and making available their model output.



This paper uses the ERA5-Land reanalysis dataset produced by Muñoz Sabater (Muñoz-Sabater et al. 2021) and was downloaded from the Copernicus Climate Change Service (C3S) Climate Data
Store: https://cds.climate.copernicus.eu/datasets/reanalysis-era5-land?tab=download.

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
