# Peer review of "disentangling the hydrological behavior of CMIP6 models"

_EGUsphere, 2024_

## Author Response (AR1)

Authors response
egusphere-2024-3225. Future changes in runoff over western and central Europe: disentangling the hydrological behavior of CMIP6 models.

We thank the reviewer for their time and constructive feedback. Their thoughtful comments and suggestions have helped us improve the clarity and overall quality of the manuscript. We appreciate the points raised and the references provided, which prompted us to re-examine important aspects of our study and strengthen both the analysis and the presentation of the results.

Our responses to the reviewers' comments (in black) are shown below in purple, and changes to the paper are shown in green. An accompanying manuscript with track changes is also provided.

**Reviewer 1**

**Summary**

In this manuscript, the authors attempt to provide insights into possible future hydrologic changes over Western and Central Europe (WCE). To do so, they make use of historical and future (ssp585) projections from CMIP6 models. They cluster models into 8 groups based on the similarity of their hydrologic responses and show that there is no consistent response across models/clusters in future annual runoff changes over WCE, with half of clusters showing strong decreases and half showing no change or modest increases. They further show that even when clusters of models show similar runoff responses, they may do so for different reasons (e.g., divergent signs/strengths of precipitation and evapotranspiration trends). Additionally, they show that it is challenging to observationally constrain the model projections owing to observational uncertainty in components of the water budget (especially P and ET trends) and a weak linkage between historical and future changes in the models. Finally, they leverage additional CMIP6 experiments from C4MIP and LS3MIP to provide a detailed look at some of the biogeophysical mechanisms driving the models' hydrological responses and find a strong role for plant physiological responses to elevated $CO_2$ and a limited role for soil moisture-atmosphere feedbacks in the spread of model responses.

**General comments**

Overall, I found the questions the authors posed around future hydrologic changes over WCE both scientifically interesting, owing to the substantial uncertainties, and societally relevant. Their analysis was thorough and technically sound. I particularly appreciated the use of multiple sets of CMIP6 experiments to provide several lines of evidence. I only have two minor concerns that I would like the authors to address before I would consider this manuscript ready for publication.

**Specific comments**

My first comment pertains to the clustering algorithm used to group models. How was the number of clusters determined? The authors do not describe the dissimilarity measure shown

on the x-axis in Figure 1 or how the dissimilarity threshold that determines the final clusters was chosen. I feel that these analytical choices do require some justification, and checking the sensitivity of the main results to the threshold/number of clusters would provide a valuable robustness check.

The number of clusters was determined empirically so that there are enough clusters to highlight different behaviors but not so many that there are single model clusters or that the diversity of behaviors is too difficult to analyze. The dissimilarity threshold is chosen accordingly to get eight clusters of models.

The Ward's linkage merges two clusters that result in the smallest increase in intra-cluster inertia. The intra-cluster inertia of a cluster C with centroid $\mu_C$ is computed as follow: $W(C) = \sum_{x \in C} ||x - \mu_C||^2$. The increase in intra-cluster inertia if clusters A and B were merged is therefore computed as follow: $\Delta W = W(A \cup B) - (W(A) + W(B))$. This increase is the dissimilarity measure that is noted on the x-axis. It can also be expressed as :

$$d(A, B) = \sqrt{|A||B|/(|A| + |B|)} \, ||\mu_A - \mu_B||^2$$

with $|A|$ and $|B|$ the number of points in clusters A and B and $||\mu_A - \mu_B||^2$ the Euclidean distance between the two cluster centroids.

The method section has been modified for more clarity on this point:

"To facilitate the analysis of the CMIP6 multi-model ensemble, the hydrological responses of the 36 models studied in this work are classified using hierarchical clustering. The hydrological response of a model is defined by its seasonal relative changes in precipitation, evapotranspiration and runoff, averaged over WCE between the periods 2081-2100 and 1995-2014 for the SSP5-8.5 scenario. When multiple members are available for one model, the multi-member ensemble mean of the changes is used. Relative changes for each season and variable are first standardized by subtracting the multi-model mean and dividing by the inter-model standard deviation. The Euclidean distance is used to compute the distance between each model based on their changes. The classification algorithm starts with 36 clusters (one per model) and then, recursively, the two closest clusters are merged, until one cluster remains. The Ward's linkage merges two clusters that result in the smallest increase in intra-cluster inertia. The dissimilarity measure is an estimation of the increase in intra-cluster inertia, it is computed as follow: $d(A, B) = \sqrt{|A||B|/(|A| + |B|)} \, ||\mu_A - \mu_B||^2$

with $|A|$ and $|B|$ the number of points in clusters A and B and $||\mu_A - \mu_B||^2$ the Euclidean distance between the two cluster centroids." (lines 118 to 128)

Below is a sensitivity analysis of the results to the number of clusters chosen. We show the clustering and the corresponding classified changes in precipitation, evapotranspiration and runoff with 4, 6 and 10 clusters (Figures R1 to R8).

The use of a limited number of clusters (4 in Figure R1 and 6 in Figure R2) in the classification of models facilitates the identification of predominant behaviors (Figures R5 and R6). We clearly see the two types of annual behaviors highlighted in the article: half of the clusters show a decrease in runoff and the other half show an increase. The clusters with a decrease in runoff either project an increase in precipitation and an even greater increase in evapotranspiration or a decrease in precipitation and small changes in evapotranspiration. We also distinguish the outlier behavior of the CCCma models, which are as dissimilar with C3 than C2 is with C1 (Figure R1) or with C1/C5 than C2 with C3/C4 (Figure R2). We could have stopped the clustering at six clusters but there are still clusters with high intra-variance (C2 and C3 in Figure R2). We have therefore added two more clusters without reducing the dissimilarity threshold too much so that TaiESM1 would not be a single model cluster.

With ten clusters, the C5 and C6 clusters are split (Figure 1) to give the new C5, C9, C6 and C10 clusters (Figure R3) with C10 being a single model cluster. In this classification, C5 and C9 have very similar behavior (their seasonal changes in R, P and E are close) except in summer. C6 and C10 also have similar behavior except for runoff in DJF and MAM with C10 showing a stronger than usual increase (Figure R6). Additionally, it is harder to identify the mechanisms that might be responsible for a particular behavior with a single model cluster (especially as there are not enough models available in the MIPs we analyzed to have at least one model per clusters). The issues are the same with 10 clusters or more.

The choice of eight clusters is therefore a good compromise between a diversity of hydrological behavior without having single model clusters or clusters with similar behaviors. Furthermore, as discussed above, having fewer or more clusters would not change the main physical interpretations of our study based on 8 clusters.

[Figure]

Figure R1: Same as Figure 1 but with four clusters of models

[Figure]

Figure R2: Same as Figure 1 but with six clusters of models

[Figure]

Figure R3: Same as Figure 1 but with ten clusters of models

[Figure]

Figure R4: Same as Figure 4 but with four clusters of models. The associated dendrogram of the clustering is given Figure R1.

[Figure]

Figure R5: Same as Figure 4 but with six clusters of models. The associated dendrogram of the clustering is given Figure R2.

[Figure]

Figure R6: Same as Figure 4 but with ten clusters of models. The associated dendrogram of the clustering is given Figure R3.

My second comment relates to a point the authors flag in their discussion on ll. 399-403: the CMIP6 models display a wide range of climate sensitivities, and as such warm by very different amounts over WCE in ssp585 by the end-of-century period the authors examine. Accordingly, it would make more analytical sense to normalize the hydrologic changes the authors examine by the temperature change over that period (i.e., %/K). This would help identify situations in which models might have a similar hydrologic response to warming, but warm different amounts, and to more effectively partition how much of the model spread is due to dynamic vs. thermodynamic differences. Additionally, this normalization would make for more interpretable expectations of future change that are benchmarked to warming levels and thus more scenario-independent.

This is a very interesting point that we thought about when writing the article. We think there are multiple ways to look at the sensitivity of hydrological changes to warming. We therefore have done three sensitivity tests.

First, we have normalized the hydrological changes by local temperature for the clusters with the same classification as in the paper, i.e. on raw hydrological changes (Figure R7). Then we have performed the classification on hydrological changes normalized by local temperature changes (Figure R8 and R9). Finally, we have performed the classification on hydrological changes normalized by ECS (Figure R10 and R11).

The normalization of hydrological changes by local temperature changes after classification (Figure R7) does not modify the two types of behaviors highlighted in Figure 4 (half of the clusters give a decrease and the other half an increase). The main discrepancies are some differences of intra-cluster variability after normalization by local temperature changes. Also, after normalizing by temperature changes, the CCCma models are still outliers but the difference between them and the rest of the models is smaller. The stronger than usual changes seen in CCCma models can therefore be attributed in part to their very large ECS, as discussed in the conclusion of the paper.

[Figure]

Figure R7: Same as Figure 4 but the changes in each season are divided by the local changes of temperature in this season. The associated dendrogram of the clustering is given Figure 4.

Overall, the classification of hydrological changes (Figure 1 in the paper) and hydrological changes normalized by local temperature changes (Figure R8) shows important similarities, but also some discrepancies. The Earth System model and high resolution climate model of CNRM/Cerfacs switches from cluster C3 of the standard classification to the clusters C7 and C8 of this new classification. Former C2 and C4 are grouped in the same cluster, therefore we can assume that the difference between their changes is of thermodynamic origin, or at least correlated with temperature. The normalization by local temperature changes also highlights the outlier behavior of CAMS-CSM1-0, which shows strong annual hydrological changes despite small increase in temperature (C4 in Figure R9 a, b, c). This model presents an unusual runoff increase in summer despite a decrease in precipitation and an increase in evapotranspiration (Figure R9 m, n ,o).

[Figure]

Figure R8: Same as Figure 1 but the hydrological changes in each season are normalized by the local temperature changes in each season.

[Figure]

Figure R9: Same as Figure 4 but the changes in each season are divided by the local changes of temperature in this season. The associated dendrogram of the clustering is given Figure R8.

Finally, we have performed the classification on the hydrological changes normalized by the ECS. This has been done using the 30 models out of the 36 for which the ECS is available in the literature.

With the classification the CCCma model is no longer an outlier because of its very large ECS (Figures R10 and R11).

[Figure]

Figure R10: Same as Figure 1 but the hydrological changes are normalized by the ECS.

[Figure]

Figure R11: Same as Figure 4 but the changes are divided by the ECS. The associated dendrogram of the clustering is given Figure R10.

In conclusion, normalizing hydrological changes by temperature changes may lead to interesting discussions, as above, some of them already in the paper, but we are not sure that it is necessarily more interesting than discussing the raw hydrological changes. In addition, it leads to methodological questions: should we normalize by ECS, local annual temperature changes, local seasonal temperature changes? This choice has implications on the results and for the interpretation of the results. Moreover, even if normalizing by temperature seemingly reduces the hydrological difference between models, we are not sure that it should be necessarily interpreted as: "thermodynamic processes, scaling with temperature (e.g. the Clausius-Clapeyron relationship) explain the difference between models". This may not be causal or the causality could even be reversed. In some cases, the spread of temperature changes may be explained by hydrological changes rather than the other way around. For example, the physiological effects of $CO_2$ may reduce transpiration and therefore increase temperature.

Technical comments

I appreciated the authors' Table 2 and related discussion about the overlap in model components in their clusters. I'd encourage the authors to take a look at and cite the "model genealogy" literature (a few recommended papers below). to contextualize and strengthen this section.

We thank the reviewer for the references. We have added some discussion concerning model genealogy in the discussion related to table 2:

"The clustering highlights the impact of the interdependency of climate models, which has been discussed in several studies (e.g. Knutti et al. 2013, Boé 2018, Kuma et al. 2023), and may lead to an underestimation of climate change uncertainties (Steinschneider et al. 2015). The climate models that share several components (Table 2) are generally in the same cluster, except for NorESM2-LM/NorESM2-MM. This is somewhat surprising, as NorESM2-LM and NorESM2-MM differ only in resolution (and a very limited number of parameters in the atmosphere component, Seland et al. (2020)). Internal variability could explain why these two very similar models belong to different clusters, especially since only one member is available for these two models. The Earth System Models (ESMs) and Coupled Models (CM) developed by the same institute (such as CNRM-CM6-1 and CNRM-ESM2-1, or CMCC-ESM2 and CMCC-CM2-SR5) are generally in the same cluster, suggesting a secondary role for the additional components included in these ESMs. Interestingly, the models from C6 all share the same land surface and/or atmosphere components (CLM and CAM, respectively). Sharing only the atmospheric component is not always sufficient for models to have similar hydrological behavior over WCE. For example, the MetOffice models (HadGEM and UKESM) are not in the same cluster as ACCESS models. Two clusters, grouping only different versions of the same model, C7 (EC-Earth-Consortium models) and C8 (CCCma models), are quite far from the rest of the models, pointing to distinct hydrological behaviors." (lines 130 to 143)

How much is the correlation between DJF SLP and precipitation change shown in Figure 6c driven by the outliers in C8 and TaiESM1? Given that all models show increased DJF precipitation regardless of the sign of SLP change, I'm not sure what conclusions to make about the importance of circulation changes for wintertime precipitation changes.

The correlation still holds but is less significant without the C8 models and TaiESM1 (Figure R12a, b, p<0.05 instead of p<0.01). However, we agree that other processes are likely to be important: the inter-model correlations with and without TaiESM1 and the C8 models are greater in magnitude for precipitation changes normalized by temperature change (Figure R12, c and d) pointing to an impact of temperature and thermodynamics processes in addition to the one of large scale circulation.

We have added the following sentence in the paper "The inter-model correlation in Fig. 6c is lower when the three outlier models (TaiESM1 and the C8 models) are removed, but is still significant (r=-0.42 with p < 0.05, not shown)." (lines 288-289)

[Figure]

Figure R12: (a) Scatterplot between winter changes in precipitation (%) averaged over WCE (x-axis) and winter changes in sea level pressure (hPa) averaged over WCE (y-axis) of all CMIP6 models. The Pearson correlation coefficient ("r") and the corresponding p-value are given in the figure. (b) Same as (a) but without C8 models nor TaiESM1. (c) and (d) same as (a) and (b) but the DJF precipitation changes are normalized by the DJF temperature changes.

Knutti, R., D. Masson, and A. Gettelman (2013), Climate model genealogy: Generation CMIP5 and how we got there, *Geophys. Res. Lett.*, 40, 1194–1199, doi:10.1002/grl.50256.

Kuma, P., Bender, F. A.-M., & Jönsson, A. R. (2023). Climate model code genealogy and its relation to climate feedbacks and sensitivity. *Journal of Advances in Modeling Earth Systems*, 15, e2022MS003588. https://doi.org/10.1029/2022MS003588

Steinschneider, S., R. McCrary, L. O. Mearns, and C. Brown (2015), The effects of climate model similarity on probabilistic climate projections and the implications for local, risk-based adaptation planning. *Geophys. Res. Lett.*, 42, 5014–5044. doi: 10.1002/2015GL064529.

**Reviewer 2**

General Comment:

This manuscript provides a comprehensive analysis of future runoff changes over western and central Europe using the CMIP6 multi-model ensemble under a high-end emissions scenario. The hierarchical classification approach used to categorize models is helpful to identify the sources of inter-model uncertainty. I also appreciate the authors investigate multiple mechanisms using other experiments such as LS3MIP and C4MIP

Specific comments:

1. Regarding the BGC simulations shown in Fig. 9, the authors did not discuss runoff results. Unlike ET and P, runoff does not show a statistically significant correlation between ALL and BGC scenarios. It also shows varied signs of change between ALL and BGC, e.g., runoff in BGC is negative while ALL is positive, whereas ET and P show more consistent signs of change. It would be valuable to discuss these discrepancies explicitly. This reference may be relevant: Lesk, Corey S., Jonathan M. Winter, and Justin S. Mankin. "Projected runoff declines from plant physiological effects on precipitation." *Nature Water* (2025): 1-11.

We thank the reviewer for the reference. We really enjoyed this article and we found it relevant for the results presented in Fig. 9. First, note that the correlation between ALL and BGC for runoff is close to be significant at a reasonable level (p<0.076). Also, for a majority of models,

the changes in runoff in BGC are negative. Our results are therefore overall consistent with Lesk et al. (2025).

We have added a discussion about the uncertainty of the runoff response to $CO_2$ physiological effect at the end of the paragraph discussing the results of Fig. 9.

"The physiological effect of $CO_2$ therefore generally leads to both a direct decrease in evapotranspiration and in an indirect decrease in precipitation, and uncertainty remains as to how the physiological effect of CO2 will ultimately affect changes in runoff. The inter-model correlation between runoff changes in BGC and ALL is less robust (Fig. 9c, r=0.66 and $0.05<p<0.1$) than for precipitation and evapotranspiration, but still 5 out of 8 models show a decrease in runoff in BGC. This is in agreement with Lesk et al. (2025), who highlight the crucial role of the response of precipitation to the physiological effect of $CO_2$ for changes in runoff. " (lines 342 to 348)

2. Is there a particular reason the authors chose a 20-year climatology period (2081-2100 vs. 1995-2014)? The standard practice is typically using a 30-year climatology, as used in the given reference above.

We have used the 20-year climatology period 2081-2100 vs 1995-2014 period to ensure consistency with the IPCC AR6 WGI (Atlas 1.3.1 in Gutiérrez et al. 2021). Climate change is rapid in the late $21^{st}$ century under the SSP5-8.5 scenario, and a 30-year period may smooth the signal too much. A 20-year period is a good compromise between being long enough to smooth internal variability but short enough to capture rapid changes. However, to evaluate the trends in Fig. 5 b, d, f, h, j, we use a 30-year period (1985-2014) in order to reduce the impact of internal variability (Maher et al. 2020).

3. One finding of the authors is that multi-model mean and inter-model standard deviation could be uninformative. It would be helpful if the authors suggest alternative metrics or practices for better representing model agreement or uncertainty in hydrological projections.

This is a topic prone to discussion as there exists multiple ways to communicate about the results of climate models projections and the associated uncertainties. In our case, we believe that the storyline approach would be a very good way to take into account the diversity of hydrological response to anthropogenic forcing for practical applications. The storyline approach consists in exploring multiple plausible future evolution of the climate (Shepherd et al. 2018). In our case, we could do this by randomly selecting one model from each cluster. The full diversity of the CMIP6 ensemble would be very well captured, while keeping a reasonably small and manageable ensemble of plausible future evolutions (Monerie et al. 2017). We have strengthened the part of the conclusion referring to this point.

"Ensemble metrics such as the multi-model mean and inter-model standard deviation based on model democracy could therefore be biased. This result also highlights the interest of thinking in terms of a few groups of models with similar responses, depicting different possible

storylines of future hydrological changes, as in this study. This approach offers a more informative perspective on the uncertainties associated with future projections and could be completed, for decision-making purposes, by integrating other relevant factors, such as socio-economic factors (Shepherd et al. 2018)." (lines 371 to 375)

4. Because the mechanisms investigated are focused on particular seasons, e.g., DJF for large-scale circulation and JJA for others, in the abstract the seasons should be explicitly mentioned to avoid confusion or overly generalized claims.

We have added in the abstract the seasons corresponding to each mechanisms investigated. The modified sentences are shown in reply to the reviewer's next comment.

5. The current phrasing of "extreme hydrological changes" in the abstract could lead to misunderstandings. I though the authors suggest a focus on extremes (such as the 99th percentile runoff values), but later it becomes clear that the authors refer to models showing unusually high responses compared to other models. Clarifying this phrasing early on in the abstract would enhance reader understanding.

We have modified the abstract concerning the mention of extreme hydrological changes. We have replaced "extreme" by "unusually large".

"Finally, we show that large-scale circulation in winter and the representation of the physiological impact of $CO_2$ in summer are important for the unusually large hydrological changes projected by some models. The soil-moisture precipitation feedback is important in summer for the multi-model ensemble mean but not for the inter-model spread." (lines 20-23)

Technical corrections:

1. Line 97: ERA5-Land should be described as a reanalysis dataset rather than an observational dataset.

We have modified the description of the datasets used for evaluation for more clarity on their nature (reanalysis for ERA5-Land, remote sensing-based modeling framework for GLEAM or observational for CRU-TS).

"Three datasets are used for the evaluation of historical simulations. The reanalysis ERA5-Land (Muñoz-Sabater et al., 2021) consists of the land component of the ERA5 reanalysis and is forced by ERA5 meteorological fields. Its enhanced resolution and increased complexity in land surface representation lead to an added value compared to ERA5 for the estimation of runoff and soil moisture, among other land surface variables (Muñoz-Sabater et al., 2021). The remote sensing-based modeling framework GLEAM (Global Land Evaporation Amsterdam Model, Miralles et al. 2011; 2025) version 4.1a, uses observations of surface net radiation, near-surface air temperature, wind speed, leaf area index and vapor pressure deficit to estimate potential evapotranspiration with the Penman's equation. The surface soil moisture from

satellite observations is assimilated into the soil profile. The potential evapotranspiration is then combined with an evaporative stress factor based on root-zone soil moisture. The evapotranspiration is finally computed as the sum of transpiration, interception, bare soil evaporation, evaporation for water bodies and evaporation for regions covered by ice and/or snow. Finally, the gridded monthly data over land from the Climatic Research Unit (CRU) time series (TS) version 4 (Harris et al., 2020) provide another estimation of potential evapotranspiration and precipitation. CRU TS is based on the interpolation of weather station observations on a 0.5° resolution grid. The CRU TS potential evapotranspiration is computed with the Penman-Monteith formula from gridded mean temperature, vapor pressure, cloud cover and climatological wind field values." (lines 98 to 111)

2. Line 94: Please clarify the temporal resolution of the model output used. Does the daily mean temperature imply PET is calculated at a daily timestep? Please specify the resolution of other variables as well.

We have added a clarification about the temporal resolution of the computation of PET.
"Potential evapotranspiration is calculated at the daily time step for the 24 CMIP6 models with the necessary data (Table 1)" (line 92)

We have also specified the temporal resolution of other variables.
"The variables considered in this study are precipitation (P), evapotranspiration (ETR), transpiration (Tran), potential evapotranspiration (PET), total runoff (R), surface soil moisture (SSM) and sea level pressure (SLP). All variables are considered at the monthly time step except for potential evapotranspiration." (lines 91-92)

3. Table 2: The caption mentions "the colors of the first column", but the first column is entirely black. It is also unclear why the cluster is not in a sorted order. It only becomes clear in Fig. 4 that sorting is based on annual runoff.

Indeed, they were first colored but the journal does not support colored tables so we removed the colors and did not change the caption. We have replaced the sentence.
"The eight clusters are identified through hierarchical clustering (see Section 2 and Figure 1)." (lines 151-152)
We have rearranged the order of the clusters in Table 2 as it does not make sense, as the reviewer said, at this stage of the study. We have also rearranged the order of the clusters in the legend of the figures 6, 8 and 9.

4. Fig. 2: The caption mentions the "multi-model ensemble mean," but line 157 says one member per model was used. Please clarify why all members per model were not averaged or specify the criteria for selecting the single member.

Thank you for highlighting this inconsistency. We chose one member per model in Figure 2 in order to be coherent between the models for the computation of the variability threshold.

Comparing a member to a threshold is not the same as comparing the mean signal to that threshold. The variability threshold is intended to account for uncertainty due to internal variability, so we didn't want to smooth it for some models and not others. The selected member is always the first on the list. We have specified it in the text where it is needed.
"for one member (the first) per model" (line 93)
"On these maps, one member (the first) per model is used." (line 170)
"The colored dots and cross correspond to individual models with a single member (the first)." (caption Figure 5, line 274)

5.  Line 159-160: This methodological explanation should be moved to the methods section. Additionally, please briefly define the "linearly detrended modern period," and clarify the period used to calculate the standard deviation. Is sigma-1yr calculated from the multi-model ensemble mean or individually from each member? This clarification also applies to lines 175-180.

We have moved the definition of the variability threshold in the methods section and have clarified how it is calculated. We consider the period 1995-2014 as the modern period. Therefore, we have replaced "modern period" by 1995-2014. "Linearly detrended" means that we have removed the linear trend. As noted above, we use one member per model for this analysis and therefore $\delta_{1yr}$ is always calculated on one member.

**"2.3 Significance of the changes**

To assess the significance of projected changes, their magnitude is compared to a variability threshold. This variability threshold $\gamma$ is defined as in Gutierrez et al. 2021 (Eq. 1) and represents the amplitude of internal variability.

$$\gamma = \sqrt{2/20} * 1.645 * \delta_{1yr}$$

where $\delta_{1yr}$ is the interannual standard deviation measured calculated over the linearly detrended 1995-2014 period." (lines 152 to 156)

We have modified all parts of the text referencing the Methods section for the definition of the variability threshold (caption Figures 2 and 3, lines 171 and 188).

6.  Line 166: How do you define significant? Is it referring to statistical significance?

By significant changes we mean changes that have a magnitude greater than the variability threshold. We believe that it is clearer now that we have defined the variability threshold and its purpose in the Methods section.

7. Fig. 5: Figures 5d and 5i seem to include only GLEAM as a reference dataset, although ERA5-Land also provides the variable "Evaporation from Vegetation Transpiration." Please clarify why ERA5-Land was not included.

ERA5-Land does indeed provide the transpiration variable. We first tested it and we found that its climatological mean was very small compared to evapotranspiration (Fig. R13) so we decided not to use it. Nevertheless, in order to respond to the points raised in your commentary, we have conducted further investigation into the reasons for this low climatological value and we have found that it was due to an issue in the dataset (section "known issues" subsection "3. Swapped value of the components of the total evapotranspiration" from the ERA5-Land documentation at https://confluence.ecmwf.int/display/CKB/ERA5-Land%3A+data+documentation#). The values corresponding to the variable "Evaporation from vegetation transpiration" are replaced by the values corresponding to the variable "Evaporation from open water surfaces excluding oceans". And the values corresponding to the variable "Evaporation from vegetation transpiration" are the values corresponding to the variable "Evaporation from bare soil". We have therefore added the right values of transpiration from ERA5-Land in Figure 5 and have modified the text mentioning Figure 5 d, i.

"The simulated trends in transpiration are significantly positive in models from C1, C7, C5 and C8, in line with the significant positive trend in GLEAM and ERA5-Land." (line 245)

"A large and quasi-generalized underestimation of climatological transpiration in climate models compared to GLEAM and ERA5-Land is found." (line 255)

[Figure]

Figure R13: Seasonal and annual evapotranspiration (1st row) and transpiration (2nd row) averaged over WCE for ERA5-Land (blue) and Gleam (orange) datasets over the period 1995-2014.

8. Line 182: Section 2b should be Section 2.2.

We have replaced it.

9. Line 260: Please check and correct the ordering of letters corresponding to the correct sub-figures.

We have corrected it.

10. Lines 253-256: There seems to be some redundancy.

It was a summary of the section's results but there were too many details. We have streamlined the sentence.
"The models with future decreases in annual runoff (C1, C7, C2 and C4, see Fig. 4a) tend to have higher present-day climatological values of evapotranspiration. In addition, the C7 models have both strong present-day trends and strong projected changes in evapotranspiration ad precipitation between 2081-2100 and 1995-2014 as shown in Fig. 4a, b." (lines 263 to 265)

11. Line 242: Please be aware of the difference of observations and reanalysis. Here the authors use "observations" to refer to GLEAM and ERA5-Land, but in line 245, they mention GLEAM has a lack of direct observations for transpiration.

We agree that the difference between observations and reanalysis should be more carefully considered. We have modified the description of GLEAM in the Data section in response to your commentary 1. We believe that the sentence that you mentioned (line XX) makes more sense with this correction.

"The remote sensing-based modeling framework Global Land Evaporation Amsterdam Model (GLEAM, Miralles et al. 2011; 2025) version 4.1a" (lines 102-103)

12. Line 250: Please provide relevant citations to support the statement.

We have added a reference.

"Moreover, transpiration depends on the stomatal conductance of the plant, which could be affected by $CO_2$ through its physiological effect (Vicente-Serrano et al. 2022)." (lines 259-260)

13. Line 269: Cite Fig. 4b as well since annual precipitation is also mentioned.

We have added Fig. 4b in the citation.
"A salient feature of the cluster's hydrological response to climate change is the very strong increase in annual and winter precipitation in C8, and in one model of the C6 cluster, TaiESM1 (Fig. 4b, h)." (line 278)

14. Fig. 7: Consider adding observational data to help assess which LS3MIP experiment better aligns with observed conditions.

We are not sure that adding GLEAM and ERA5 is sufficient to assess which LS3MIP experiment better aligns with observed conditions, as large uncertainties exist and the simulations do not diverge much on the historical period (Figure R14).

[Figure]

Figure R14: as Figure 7 but with Gleam, ERA5-Land and Cru-TS.

15. Lines 298-299: The text refers to differences in ET and P between two LS3MIP experiments, but Fig. 8 shows differences between historical and SSP585 simulations.

Figure 8 shows both the differences of evapotranspiration (a) / precipitation (b) changes between historical and SSP585 simulations (y-axis) and differences of changes of evapotranspiration or precipitation between the two LS3MIP experiment divided by the difference of changes in soil moisture in the two experiments (x-axis). This metric captures the strength of the soil moisture atmosphere feedback in each model. Figure. 8 shows that there is no relationship across models between the magnitude of summer evapotranspiration (a) or precipitation (b) changes and the strength of the soil moisture - atmosphere feedback.

16. Lines 321-322: The authors mention "ALL" and "BGC", but in the figure they didn't use the same notation.

We have modified the labels of the axis in Figure 9 to maintain consistency throughout the article. We have also replaced "ssp585" by "ALL" in Figure 8.

17. Fig. 9: The caption does not follow the sub-figures being shown. The y-axis and x-axis are also not consistent with the description.

Thank you for pointing it out, the axes of the figure were inverted. We have corrected this mistake.

Gutiérrez, J.M., R.G. Jones, G.T. Narisma, L.M. Alves, M. Amjad, I.V. Gorodetskaya, M. Grose, N.A.B. Klutse, S. Krakovska, J. Li, D. Martínez-Castro, L.O. Mearns, S.H. Mernild, T. Ngo-Duc, B. van den Hurk, and J.-H. Yoon, 2021: Atlas. In *Climate Change 2021: The Physical Science Basis. Contribution of Working Group I to the Sixth Assessment Report of the Intergovernmental Panel on Climate Change* [Masson-Delmotte, V., P. Zhai, A. Pirani, S.L. Connors, C. Péan, S. Berger, N. Caud, Y. Chen, L. Goldfarb, M.I. Gomis, M. Huang, K. Leitzell, E. Lonnoy, J.B.R. Matthews, T.K. Maycock, T. Waterfield, O. Yelekçi, R. Yu, and B. Zhou (eds.)]. Cambridge University Press, Cambridge, United Kingdom and New York, NY, USA, pp. 1927–2058, doi: 10.1017/9781009157896.021.

Maher, N., Lehner, F., Marotzke, J.: Quantifying the role of internal variability in the temperature we expect to observe in the coming decades, Environ. Res. Lett., 15, 054014, 10.1088/1748-9326/ab7d02, 2020.

Monerie, P. A., Sanchez-Gomez, E. and Boé, J.: On the range of future Sahel precipitation projections and the selection of a sub-sample of CMIP5 models for impact studies, Clim. Dyn., 48, 2751–2770, https://doi.org/10.1007/s00382-016-3236-y, 2017.

Shepherd, T.G., Boyd, E., Calel, R.A. et al.: Storylines: an alternative approach to representing uncertainty in physical aspects of climate change, Climatic Change, 151, 555–571, https://doi.org/10.1007/s10584-018-2317-9, 2018.

---

## Author Response (AR2)

Dear Editor,

As you suggested, we have included in the supplementary materials several of the analyses that supported our responses to the reviewers: the sensitivity of the analysis to the choice of the number of clusters (S1), to the influence of the wide range of climate sensitivities displayed by the CMIP6 models (S2) and also the sensitivity of the inter-model relationship between SLP changes and precipitation changes in winter to the outlier behavior of some models (Fig. S12).

We have added references to these supplementary materials in the manuscript (shown in bold).

line 130: "eight clusters are finally defined empirically (Fig. 1**, see section S1 for a sensitivity analysis to the number of clusters chosen**)"

line 289: "(r=-0.42 with p < 0.05, **Fig. S12**)"

line 420: "Selecting only models within the likely range of ECS would therefore lead to the exclusion of models with atypical hydrological behaviors over WCE **(see section S2)**"

Sincerely,
Julien Boé & Juliette Deman

---

## Author Response (AR3)

Dear editorial team,

We have modified the fontsize in figures 2, 3, 7 and 8 and added back the colorbar on figure 2.
We have modified the units on the second column of figure 5. We have replaced « No units » by « $(10y)^{-1}$ » as it represents decadal trends over the period 1985-2014 that are standardised by the climatological mean over the period 1995-2014. We also modified the legend of this figure relative to this change.
We have modified the units on figures 3, 5, 7 and 8 and the legend of these figures relative to these changes. We have replaced « mm day$^{-1}$ » by « mm d$^{-1}$ » and « day$^{-1}$ » by « d$^{-1}$ ».

Sincerely,
Julien Boé & Juliette Deman